# Is Reflex Germline *BRCA1/2* Testing Necessary in Women Diagnosed with Non-Mucinous High-Grade Epithelial Ovarian Cancer Aged 80 Years or Older?

**DOI:** 10.3390/cancers15030730

**Published:** 2023-01-25

**Authors:** Robert D. Morgan, George J. Burghel, Nicola Flaum, Michael Bulman, Philip Smith, Andrew R. Clamp, Jurjees Hasan, Claire L. Mitchell, Zena Salih, Emma R. Woodward, Fiona Lalloo, Emma J. Crosbie, Richard J. Edmondson, Helene Schlecht, Gordon C. Jayson, D. Gareth R. Evans

**Affiliations:** 1Department of Medical Oncology, The Christie NHS Foundation Trust, Wilmslow Road, Manchester M20 4BX, UK; 2Division of Cancer Sciences, Faculty of Biology, Medicine and Health, University of Manchester, Manchester M13 9PL, UK; 3Manchester Centre for Genomic Medicine, North West Genomic Laboratory Hub, Saint Mary’s Hospital, Oxford Road, Manchester M13 9WL, UK; 4Division of Evolution and Genomic Sciences, Faculty of Biology, Medicine and Health, University of Manchester, Manchester M13 9PL, UK; 5Department of Clinical Genetics, Saint Mary’s Hospital, Oxford Road, Manchester M13 9WL, UK; 6Department of Gynaecological Oncology, Saint Mary’s Hospital, Oxford Road, Manchester M13 9WL, UK

**Keywords:** epithelial ovarian cancer, germline, somatic, *BRCA1*, *BRCA2*

## Abstract

**Simple Summary:**

Approximately 15% of patients diagnosed with high-grade non-mucinous epithelial ovarian cancer (EOC) have a germline *BRCA1/2* mutation, although all patients are often able to access germline testing. Importantly, the risk of familial ovarian cancer reduces with advancing age at diagnosis. The aim of our study was to determine the prevalence of germline and somatic *BRCA1/2* mutations in women diagnosed with non-mucinous high-grade EOC aged ≥80. We found that somatic *BRCA1/2* mutations occurred nine times more frequently than germline *BRCA1/2* mutations in women aged ≥80. The only germline *BRCA1/2* mutation reported in a patient aged ≥80 was detected in both germline and tumour DNA. These data suggest that germline *BRCA1/2* testing in women diagnosed with high-grade non-mucinous EOC aged ≥80 can be reserved for those with a detectable tumour *BRCA1/2* mutation.

**Abstract:**

Women diagnosed with non-mucinous high-grade epithelial ovarian cancer (EOC) in England are often reflex-tested for germline and tumour *BRCA1*/*2* variants. The value of germline *BRCA1/2* testing in women diagnosed aged ≥80 is questionable. We performed an observational study of all women diagnosed with non-mucinous high-grade EOC who underwent germline and tumour *BRCA1/2* testing by the North West of England Genomic Laboratory Hub. A subgroup of women also underwent germline testing using a panel of homologous recombination repair (HRR) genes and/or tumour testing for homologous recombination deficiency (HRD) using Myriad’s myChoice® companion diagnostic. Seven-hundred-two patients successfully underwent both germline and tumour *BRCA1/2* testing. Of these, 48 were diagnosed with non-mucinous high-grade EOC aged ≥80. In this age group, somatic *BRCA1/2* pathogenic/likely pathogenic variants (PV/LPVs) were detected nine times more often than germline *BRCA1/2* PV/LPVs. The only germline PV reported in a patient aged ≥80 was detected in germline and tumour DNA (*BRCA2* c.4478_4481del). No patient aged ≥80 had a germline PV/LPVs in a non-*BRCA1/2* HRR gene. Thirty-eight percent of patients aged ≥80 had a tumour positive for HRD. Our data suggest that tumour *BRCA1/2* and HRD testing is adequate for patients diagnosed with non-mucinous high-grade EOC aged ≥80, with germline *BRCA1/2* testing reserved for women with a tumour *BRCA1/2* PV/LPVs.

## 1. Introduction

Inclusion of poly (ADP-ribose) polymerase inhibitors (PARPi) as standard therapy for non-mucinous high-grade epithelial ovarian cancer (EOC) led to mainstream, reflex germline and tumour *BRCA1/2* testing [1]. Prior to the era of PARPi, index cases of ovarian cancer were selected for germline *BRCA1/2* testing based on their risk of hereditary cancer, with age at diagnosis and family history used to determine risk [2,3,4]. The present lack of selection criteria for germline *BRCA1/2* testing means many women at low risk of being a germline heterozygote, especially elderly women, undergo germline testing unnecessarily [5,6,7,8,9,10,11].

The range of genetic tests available for women diagnosed with ovarian cancer is expanding. In England, national guidelines specify that all women with newly diagnosed high-grade ovarian cancer undergo tumour testing for *BRCA1/2* pathogenic/likely pathogenic variants and homologous recombination deficiency (HRD). National guidelines also specify that all women diagnosed with non-mucinous high-grade EOC undergo germline testing for genes associated with familial ovarian cancer regardless of age. These cancer predisposition genes include *BRCA1*, *BRCA2*, *BRIP1*, *PALB2*, *RAD51C*, *RAD51D*, *MLH1*, *MSH2* and *MSH6* [12]. This strategy of germline testing all unselected cases of ovarian cancer provides an opportunity to determine prevalence rates across age groups, thereby optimizing future testing pathways [13].

In this study, we report the prevalence of pathogenic/likely pathogenic variants (hereon referred to as ‘pathogenic variants’) in ovarian cancer susceptibility genes in a large group of unselected women diagnosed with non-mucinous high-grade EOC. In addition, we report the prevalence of somatic *BRCA1/2* pathogenic variants and tumours positive for HRD. Our aim was to define the optimal germline and tumour testing strategy for women diagnosed with non-mucinous high-grade EOC aged ≥80.

## 2. Materials and Methods

Unselected women were included who had been diagnosed with non-mucinous high-grade carcinoma of the ovary, fallopian tube or primary peritoneum who successfully underwent both germline and tumour *BRCA1/2* testing by the Genomic Laboratory Hub in the North West of England between 1 May 2016 and 15 November 2022. Eligible histological subtypes included high-grade serous, high-grade endometrioid (moderately differentiated (grade 2) or poorly differentiated (grade 3)), clear cell and poorly differentiated adenocarcinoma not otherwise specified [14]. Cases of ovarian carcinosarcoma were excluded. All FIGO (International Federation of Gynecology and Obstetrics) stages of ovarian cancer were eligible for inclusion [15]. All index cases that failed germline or tumour *BRCA1/2* testing were excluded.

The next-generation sequencing (NGS) and multiplex ligation-dependent probe amplification (MLPA) assays used to test for germline *BRCA1/2* pathogenic variants have been described previously [16]. From 1 April 2022, germline DNA from index cases also routinely underwent multi-gene panel testing for pathogenic variants in *BRCA1*, *BRCA2*, *BRIP1*, *PALB2*, *RAD51C*, *RAD51D*, *MLH1*, *MSH2* and *MSH6* [12]. The genes associated with homologous recombination repair (HRR) included *BRCA1*, *BRCA2*, *BRIP1*, *PALB2*, *RAD51C* and *RAD51D*. The NGS enrichment method used a custom-designed Agilent SureSelect^TM^ panel, including the coding region of transcripts and splice sites +/- 15 base pairs and known intronic pathogenic variants. Following NGS enrichment, samples were sequenced using the Illumina NextSeq 550 System. The overall coverage had to be >99% at 100X to pass the sample. Single nucleotide variants and small deletions, duplications, insertions, and insertion/deletions (<40 base pairs) were called using an in-house custom bioinformatic analysis pipeline validated to detect heterozygous and mosaic variants to a variant allele frequency (VAF) of ≥4%. The reference sequences used included NM_007294.3 (BRCA1), NM_000059.3 (BRCA2), NM_032043.2 (BRIP1), NM_024675.3 (PALB2), NM_058216.1 (RAD51C), NM_002878.3 (RAD51D), NM_001142571.1 (RAD51D), NM_000249.3 (MLH1), NM_000251.1 (MSH2) and NM_000179.2 (MSH6). Sequence variant nomenclature followed Human Genome Variation Society (HGVS) guidelines (http://varnomen.hgvs.org accessed on 16 November 2022). Sequence variants were classified according to the Association for Clinical Genomic Science (ACGS) Best Practice Guidelines [17].

The NGS assay used to test for tumour *BRCA1/2* pathogenic variants has been described previously [18]. From 11 April 2021, tumour DNA from index cases was also routinely tested using Myriad’s myChoice^®^ companion diagnostic (CDx) [19]. Myriad Genetics, Inc. (Salt Lake City, UT, USA) performed the myChoice^®^ CDx. The eligibility criteria for myChoice^®^ CDx testing included newly diagnosed FIGO stage III/IV high-grade epithelial ovarian, fallopian tube or primary peritoneal cancer. The myChoice^®^ CDx reported *BRCA1/2* pathogenic variants and a genomic instability score (GIS) in tumour DNA. The GIS was a composite score of three bioinformatic algorithms that detected putative biomarkers of HRD, including loss of heterozygosity (LOH), telomeric allelic imbalance (TAI) and large-scale state transitions (LST) [20,21,22]. A GIS of ≥42 was reported as GIS-positive, while a GIS of <42 was reported as GIS-negative. Any tumour with a *BRCA1/2* pathogenic variant or a GIS of ≥42 was reported as HRD-positive. Any tumour with *BRCA1/2* wild type and a GIS of <42 was reported as HRD-negative.

Categorical data were reported as number and percentage. Continuous data were reported as mean and range. For categorical data, the chi-squared test was used to determine if significant differences occurred between groups. For continuous data, Student’s *t*-test was used to determine if significant differences occurred between the means of two groups.

## 3. Results

Seven-hundred-two women diagnosed with non-mucinous high-grade EOC underwent germline and tumour *BRCA1/2* testing (Table 1). Seventy-five (11%) and fifty-four (8%) women were found to have a germline or somatic *BRCA1/2* pathogenic variant, respectively (Table 1). Most *BRCA1/2* pathogenic variants were detected in women diagnosed with high-grade serous carcinoma (110/129; 85%), although this was the commonest histological subtype tested (623/702; 89%) (Table 1).

The concordance between germline *BRCA1/2* pathogenic variants detected in germline and tumour DNA was 94% (44/47) and 100% (28/28) using our in-house tumour *BRCA1/2* assay and Myriad’s myChoice^®^ CDx, respectively. The three germline *BRCA1/2* variants missed using our in-house tumour *BRCA1/2* assay included *BRCA1* Exon 13 duplication, *BRCA2* Exon 1–2 deletion and *BRCA2* Exon 14–16 deletion. Three germline *BRCA1/2* large genomic rearrangements were detected in germline and tumour DNA using Myriad’s myChoice^®^ CDx, including *BRCA1* Exon 9-12 deletion (GIS 56), *BRCA1* Exon 17 deletion (GIS 70) and *BRCA2* Exon 14-16 deletion (GIS 36).

By categorizing germline and somatic *BRCA1/2* pathogenic variants according to age, it was clear that no patient diagnosed aged ≥80 had a germline *BRCA1* pathogenic variant (Table 2). The only germline *BRCA2* pathogenic variant reported in a patient diagnosed aged ≥80 was detected in germline and tumour DNA (*BRCA2* c.4478_4481del; GIS 83). Patients diagnosed aged ≥80 were nine times more likely to have a somatic versus germline *BRCA1/2* pathogenic variant (Table 2). The likelihood of having a germline versus somatic *BRCA1/2* pathogenic variant reduced with each decade of age at diagnosis ≥60 (Figure 1). These data suggest that routine germline *BRCA1/2* testing is unnecessary in women diagnosed with non-mucinous high-grade EOC aged ≥80.

We next categorized GIS status according to age at diagnosis to determine whether genomic instability testing was necessary in women diagnosed with non-mucinous high-grade EOC aged ≥80. Of the 702 patients included in this study, 346 (49%) had been tested using Myriad’s myChoice^®^ CDx (Table 3). Over 90% of the patients tested had been diagnosed with high-grade serous or high-grade endometrioid carcinoma (320/346; 92%) (Table 3). The mean age of patients with a GIS-positive versus GIS-negative tumour differed significantly (62.7 versus 66.0 years; *p* = 0.004) (Figure 2). For those patients aged ≥80, the likelihood of having a GIS-positive versus GIS-negative tumour was similar (Table 4). There was little variation in the likelihood of having a GIS-positive versus GIS-negative tumour across age groups (Figure 3). These data support the use of tumour *BRCA1/2* and genomic instability score testing in women diagnosed with non-mucinous high-grade EOC aged ≥80.

Finally, we investigated the prevalence of germline pathogenic variants in non-*BRCA1/2* HRR-associated genes in women diagnosed with non-mucinous high-grade EOC aged ≥80 who also underwent tumour testing using Myriad’s myChoice^®^ CDx. Of the 702 patients included in this study, 174 had undergone germline multi-gene panel testing since 1 April 2022. In the 119 patients with germline and somatic *BRCA1/2* wild type, four germline *BRIP1* pathogenic variants were detected (Table 5). No other germline HRR genes were detected in this cohort. All four germline *BRIP1* pathogenic variants were detected in women aged <70 at diagnosis with at least one first- or second-degree relative diagnosed with breast or ovarian cancer (Table 6). Two of the four *BRIP1* pathogenic variants were detected in women with a GIS-positive tumour (Table 6). These data provide limited support for the use of germline multi-gene panel testing in women diagnosed aged ≥80, regardless of GIS status.

## 4. Discussion

Germline *BRCA1/2* pathogenic variants are predictive and prognostic biomarkers in ovarian cancer. *BRCA1/2*-mutant ovarian tumours are highly sensitive to DNA damaging agents, such as platinum chemotherapy and PARPi [23,24,25,26,27]. Women diagnosed with germline *BRCA1/2*-mutant EOC have improved survival outcomes compared to sporadic cases [28]. More broadly, identifying germline *BRCA1/2* pathogenic variants leads to cascade testing and risk-reducing strategies in related, unaffected germline heterozygotes [29,30]. Thus, detection of germline *BRCA1/2* pathogenic variants is important for index cases and their family. However, germline *BRCA1/2* pathogenic variants occur in only around 15% of EOC, meaning many patients, especially elderly women, undergo unnecessary testing [31].

Opinions differ regarding whether or not multi-gene germline and somatic *BRCA1/2* testing is cost-effective for accessing PARPi therapy [32,33,34]. To reduce overall costs, multi-disciplinary teams could omit reflex germline testing in women at low risk of germline *BRCA1/2* heterozygotes. Data from our study show that women diagnosed with non-mucinous high-grade EOC aged ≥80 can be considered ‘very low risk’ and, therefore, do not require reflex germline *BRCA1/2* testing. Instead, germline testing could be reserved for women who have a *BRCA1/2* pathogenic variant detected first in tumour DNA, or when tumour testing fails. Based on our data, germline *BRCA1/2* testing in women aged ≥80 would be reduced by around 80% (10/48 patients would have required testing in our cohort). In the United Kingdom, over 1000 women are diagnosed each year with non-mucinous high-grade EOC aged ≥80, meaning an estimated 800 cases could avoid germline testing [35]. The multi-gene germline panel NGS test used in the North West Genomic Laboratory Hub costs GBP 550 per test. If testing costs are similar across the United Kingdom, we estimate cost savings of GBP 440,000 each year by omitting reflex germline testing in women aged ≥80.

The limitation of relying upon upfront tumour *BRCA1/2* testing to identify all possible pathogenic variants is that most local tumour NGS assays are not yet validated to detect whole gene/exon deletions or duplications [11,36,37], although Myriad’s myChoice^®^ CDx does reliably detect these variants [19]. Large genomic rearrangements account for <10% of all germline *BRCA1/2* pathogenic variants [38]. In our study, all 8% of germline large genomic rearrangements occurred in women aged <70. The only patient diagnosed with a germline *BRCA1/2* pathogenic variant aged ≥80 had a small deletion (four base pairs in size) in *BRCA2* that was detected first in tumour DNA and then subsequently in germline DNA. These data support a strategy of performing only upfront tumour *BRCA1/2* testing in women aged ≥80.

Identifying women with newly diagnosed advanced high-grade EOC and a GIS-positive tumour expands access to first-line olaparib plus bevacizumab maintenance therapy [39]. In addition, women with HRD-positive tests have better survival outcomes following PARPi maintenance monotherapy compared to those with HRD-negative tests [40,41,42,43,44]. Our study shows that genomic instability/mutational ‘scarring’ is not an age-defined biomarker. Indeed, in our cohort, there was little variation in the likelihood of an index case having a GIS-positive versus GIS-negative tumour across age groups. These data support use of HRD tumour testing in unselected cases of non-mucinous high-grade EOC regardless of age.

To assess the correlation between GIS-positive tumours and germline pathogenic variants in HRR genes, we analysed a subgroup of patients that were tested using Myriad’s myChoice^®^ CDx and a standard panel of ovarian cancer susceptibility genes [12]. Unsurprisingly, germline *BRCA1/2* pathogenic variants were the most frequently detected HRR mutations in women with GIS-positive tumours. The low prevalence rate of germline pathogenic variants in other HRR genes was predictable due to the lack of selection criteria used for HRR testing [45,46,47]. The absence of germline pathogenic variants in non-*BRCA1/2* HRR genes in women aged ≥80, combined with the fact that only 50% of these variants were found in women with a GIS-positive tumour, suggest germline multi-gene panel testing has limited value in women aged ≥80 regardless of GIS status. Indeed, in all patients tested, the addition of four HRR genes brought about a very small uplift in the potential causes of HRD in *BRCA1/2* wild type/GIS-positive tumours, with only 2/35 additional pathogenic variants identified. We recognize that the panel of genes used in our study includes only commonly tested ovarian cancer susceptibility genes, therefore not accounting for alternative genetic and epigenetic drivers of HRD [48,49,50,51].

It is unclear why two of the four patients with a germline *BRIP1* pathogenic variant had a GIS-negative tumour. By contrast, only 3% (1/31) of patients with a germline *BRCA1/2* pathogenic variant had a GIS-negative tumour (*p* = 0.002). One explanation could be the absence of biallelic inactivation of the *BRIP1* gene, meaning such tumours were not BRIP1-protein-deficient and, therefore, HRR proficient [52]. This finding raises the question as to whether HRD-test-negative tumours containing a non-*BRCA1/2* HRR pathogenic variant will respond to PARPi if they are in fact HRR proficient [53,54,55,56,57,58]. This will require close monitoring for poorer responses to PARPi in women with lower-penetrance HRR genes, in which ovarian cancers may have occurred sporadically. Interrogating tumour DNA for gene-specific loss of heterozygosity in germline carriers of non-*BRCA1/2* HRR pathogenic variants may help to clarify whether borderline GIS-negative tumours have biallelic inactivation and are more likely to respond to PARPi [52].

There are two limitations with this study. Firstly, genetic testing practices varied during the study period. Thus, information on germline multi-gene panel and GIS testing was only available for a subgroup of patients. Secondly, the number of patients tested for non-*BRCA1/2* HRR genes was relatively small when considering the population frequency of these moderate-to-low-penetrance genes. These two limitations make it difficult to draw any conclusions regarding age distribution and certainty, in order to make changes to policy recommendations.

## 5. Conclusions

We provide evidence demonstrating that the age-based threshold for reflex, mainstream germline *BRCA1/2* testing in unselected women diagnosed with non-mucinous high-grade EOC could be set at <80 years old. In this age group, it may be more appropriate to focus resources on reflex *BRCA1/2* and HRD tumour testing, with confirmatory germline *BRCA1/2* testing reserved for those patients with a tumour-pathogenic variant.

## Figures and Tables

**Figure 1 cancers-15-00730-f001:**
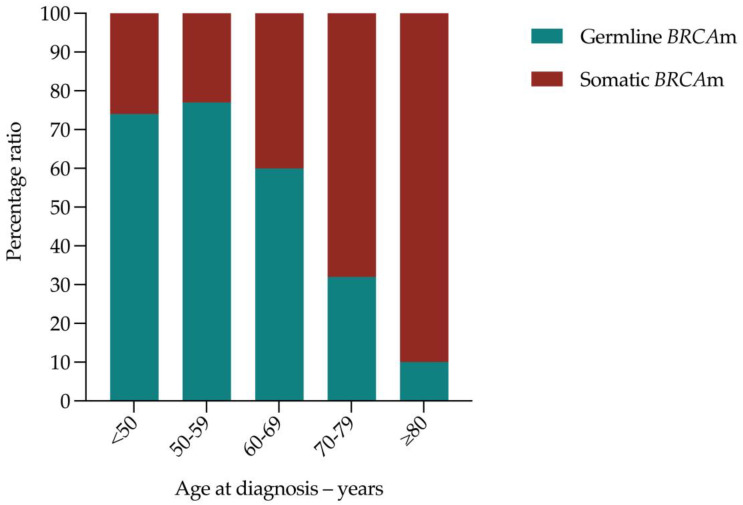
Percentage ratio of germline versus somatic *BRCA1/2* pathogenic variants according to age group. Key: *BRCA*m, *BRCA1/2* pathogenic variant.

**Figure 2 cancers-15-00730-f002:**
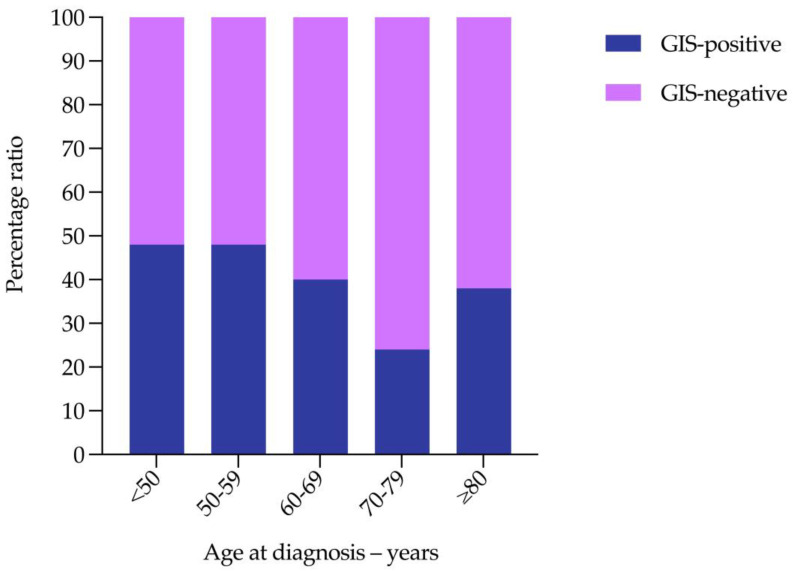
Percentage ratio of GIS-positive versus GIS-negative tumours according to age group.

**Figure 3 cancers-15-00730-f003:**
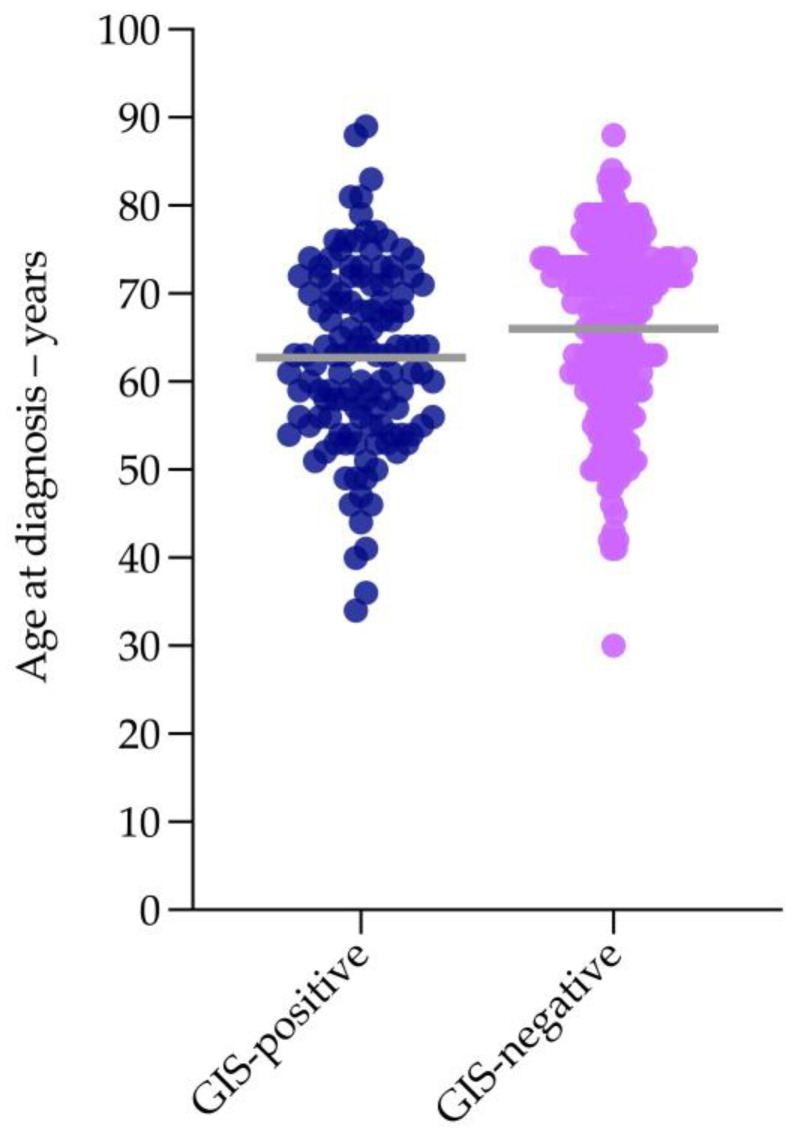
Distribution of age in GIS-positive and GIS-negative tumours. Key: each dot represents the age of an individual patient; grey bar represents mean average.

**Table 1 cancers-15-00730-t001:** Demographic data for patients that underwent *BRCA1/2* testing. Data are presented as mean (range) or number (percentage; the denominator being the ‘Total number of patients tested’). Key: *BRCA*m, *BRCA1/2* pathogenic variant; *BRCA*wt, *BRCA1/2* wild type; NOS, not otherwise specified.

Demographic	Number of Patients Tested	*BRCA*m		*BRCA*wt
		Germline	Somatic	
**Age at diagnosis—years**	64 (20–92)	58 (35–80)	67 (40–92)	65 (20–89)
**Histology**				
High-grade serous	623 (89)	61	49	513
High-grade endometrioid	34 (5)	2	2	30
Clear cell	32 (5)	4	3	25
Adenocarcinoma, NOS	11 (2)	8	0	3
Mixed	2 (<1)	0	0	2
Total	702	75 (11)	54 (8)	573

**Table 2 cancers-15-00730-t002:** Germline and somatic *BRCA1/2* pathogenic variants categorized by age group. Data are presented as number (percentage; the denominator is the ‘Number of patients tested’ in each age group). Key: *BRCA*m, *BRCA1/2* pathogenic variant.

Age at Diagnosis—Years	Number of Patients Tested	Germline Mutations			Somatic Mutations			% Total Germline to Somatic *BRCA*m Ratio
		*BRCA1*	*BRCA2*	Total	*BRCA1*	*BRCA2*	Total	
<50	75	12	2	14 (19)	4	1	5 (7)	19:7
50–59	168	18	16	34 (20)	7	3	10 (6)	10:3
60–69	207	5	14	19 (9)	9	6	15 (7)	9:7
70–79	204	1	6	7 (3)	7	8	15 (7)	3:7
≥80	48	0	1	1 (2)	3	6	9 (19)	1:9
Total	702	36	39	75 (11)	30	24	54 (8)	11:8

**Table 3 cancers-15-00730-t003:** Demographic data for patients that were tested using Myriad’s myChoice^®^ CDx. Data are presented as mean (range) or number (percentage; the denominator being the ‘Total’). Key: *BRCA*m, *BRCA1/2* pathogenic variant; *BRCA*wt, *BRCA1/2* wild type; GIS-, GIS-negative; GIS+, GIS-positive; NOS, not otherwise specified. The four *BRCA*m/GIS- tumors included germline *BRCA2* Exon 14-16 deletion (GIS 36), somatic *BRCA1* Exon 13-24 deletion (GIS 34), somatic *BRCA2* c.9097dup (GIS 5) and somatic *BRCA2* c.3760G>T (GIS 24).

Demographic	Number of Patients Tested	HRD-Negative Tumours	HRD-Positive Tumours			
			*BRCA*m/GIS+	*BRCA*m/GIS-	*BRCA*wt/GIS+	Total
**Age at Diagnosis—Years**	65 (30–89)	66 (30–88)	63 (41–88)	64 (53–75)	63 (34–89)	63 (34–89)
**Histology**						
High-grade serous	292 (84)	183	42	2	65	109
High-grade endometrioid	28 (8)	18	0	2	8	10
Clear cell	22 (6)	13	5	0	4	9
Adenocarcinoma, NOS	2 (<1)	0	1	0	1	2
Mixed	2 (<1)	2	0	0	0	0
Total	346	216	48	4	78	130

**Table 4 cancers-15-00730-t004:** Genomic instability score status categorized by age group. Data are presented as number (percentage; the denominator is the ‘Number of patients tested’ in each age group). Key: *BRCA*m, *BRCA1/2* pathogenic variant; *BRCA*wt, *BRCA1/2* wild type.

Age at Diagnosis—Years	Number of Patients Tested	GIS-Positive			GIS-Negative			% Total GIS-Positive to GIS-Negative Ratio
		*BRCA*m	*BRCA*wt	Total	*BRCA*m	*BRCA*wt	Total	
<50	23	2	9	11 (48)	0	12	12 (52)	12:13
50–59	84	20	20	40 (48)	2	42	44 (52)	12:13
60–69	98	15	24	39 (40)	0	59	59 (60)	2:3
70–79	128	8	23	31 (24)	2	95	97 (76)	6:19
≥80	13	3	2	5 (38)	0	8	8 (62)	19:31
Total	346	48	78	126 (36)	4	216	220 (64)	9:16

**Table 5 cancers-15-00730-t005:** Homologous recombination repair genes categorized by genomic instability score status. Data are presented as number (percentage; the denominator being the ‘Total *BRCA*wt’). Key: ^a^
*BRCA2* Exon 14-16 deletion (GIS 36); ^b^
*BRCA1* Exon 13-24 deletion (GIS 34); ^c^
*BRCA2* c.9097dup (GIS 5) and *BRCA2* c.3760G>T (GIS 24); *BRCA*m, *BRCA1/2* pathogenic variant; *BRCA*wt, *BRCA1/2* wild type; HRRm, homologous recombination repair pathogenic variant.

GIS Status	Number of Patients Tested	Germline		Somatic		*BRCA*wt	Germline Non-*BRCA1/2* HRRm			
		*BRCA1*	*BRCA2*	*BRCA1*	*BRCA2*		*BRIP1*	*PALB2*	*RAD51C*	*RAD51D*
GIS-positive	86	14	16	12	9	35	2	0	0	0
GIS-negative	88	0	1 ^a^	1 ^b^	2 ^c^	84	2	0	0	0
Total	174	14	17	13	11	119	4 (3)	0	0	0

**Table 6 cancers-15-00730-t006:** Demographic data for patients diagnosed with a germline *BRIP1* pathogenic variant. Key: *BRCA*m, *BRCA1/2* pathogenic variant; FDR, first-degree relative; GIS, genomic instability score; HGSOC, high-grade serous ovarian cancer; SDR, second-degree relative; WT, wild type.

*BRIP1* Variant		Age at Diagnosis—Years	Histology	FIGO Stage	Family History	*BRCA*m		GIS	HRD status
Nucleotide Level	Protein Level					Germline	Tumour		
c.1888dup	(p.Thr630fs)	68	HGSOC	IIIC	1 × FDR Breast Cancer	WT	WT	40	Negative
c.2108delinsTCC	(p.Lys703fs)	68	HGSOC	IIIC	1 × FDR Ovarian Cancer	WT	WT	49	Positive
c.2392C>T	(p.Arg798Ter)	59	HGSOC	IVB	1 × SDR Breast Cancer	WT	WT	57	Positive
c.2492+2dup	p.(?)	60	HGSOC	IVA	3 × FDR Breast Cancer	WT	WT	41	Negative

## Data Availability

The data presented in this study are available on request from the corresponding authors.

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
