# Peer review of "Is Reflex Germline BRCA1/2 Testing Necessary in Women Diagnosed with Non-Mucinous High-Grade Epithelial Ovarian Cancer Aged 80 Years or Older?"

_cancers, 2023, doi:10.3390/cancers15030730_

Round 1
Reviewer 1 Report
Authors present observational data from (consecutive?) germline and tumour testing of high-grade non-mucinous ovarian cancers.
Some information could be added to enhance the findings and support the conclusions:
Please define the study period.
Are these patients unselected consecutive patients who fit the inclusion criteria for testing? what proportion of high grade ovarian cancer patients in the catchment area of this laboratory do they represent?
Authors state that only one patient 80 years and older had a germline mutation in BRCA, and that this mutation was also detected by somatic testing. How many patients overall had germline mutations that were not detected by somatic testing?
How many patients had panel germline testing overall?
Limitations are not highlighted or discussed. For example:
Genetic testing practices varied over the course of the study period. This means that information on germline panel testing and GIS/HRD testing was available for a subgroup only.
Non-BRCA germline pathogenic mutations are uncommon and the number of these in the study cohort tested is very small. This makes it difficult to draw any conclusions re age distribution and certainly, to make any policy recommendations.
An argument could be made for the use of somatic testing to select ovarian cancer cases for germline testing across the board. Please discuss why upfront somatic testing using an assay that includes LGRs is not cost effective (as opposed to using more limited NGS assays selectively in different age groups).
Author Response
Responses are provided in bold
Reviewer 1
Authors present observational data from (consecutive?) germline and tumour testing of high-grade non-mucinous ovarian cancers.
Some information could be added to enhance the findings and support the conclusions:
Please define the study period.
The study period was between 1st May 2016 and 15th November 2022. We have added this information to the revised manuscript.
Are these patients unselected consecutive patients who fit the inclusion criteria for testing? what proportion of high grade ovarian cancer patients in the catchment area of this laboratory do they represent?
Yes, unselected, but not consecutive because we have included all patients that have successfully undergone germline and tumour BRCA1/2 testing, and not all patients diagnosed with epithelial ovarian cancer during the study time period would have been tested.
The Manchester Cancer Genomic Medicine laboratory receives germline and tumour BRCA1/2 testing requests from many cancer centres throughout England and Northern Ireland. We are therefore unable to speculate the proportion of high-grade non-mucinous EOC cases our data comprises.
Authors state that only one patient 80 years and older had a germline mutation in BRCA, and that this mutation was also detected by somatic testing. How many patients overall had germline mutations that were not detected by somatic testing?
Two germline BRCA1/2 pathogenic variants were missed in tumour DNA. Both were large genomic rearrangements (LGRs). Both LGRs were missed using our local NGS based tumour BRCA1/2 testing assay and not Myriad’s myChoice® HRD testing assay. The concordance between germline and tumour LGRs with Myriad’s myChoice® HRD testing assay was 100%. This test only became available for use in April 2021.
How many patients had panel germline testing overall?
Germline multi-panel testing was performed on 174 patients. This information is available in Table 5. We have now included an additional sentence in the results section of the revised manuscript stating this.
Limitations are not highlighted or discussed. For example:
Genetic testing practices varied over the course of the study period. This means that information on germline panel testing and GIS/HRD testing was available for a subgroup only.
Non-BRCA germline pathogenic mutations are uncommon and the number of these in the study cohort tested is very small. This makes it difficult to draw any conclusions re age distribution and certainly, to make any policy recommendations.
We have now included these limitations in the revised manuscript. We have also toned down the conclusion to account for these limitations. We are grateful for the Reviewer’s suggestions.
An argument could be made for the use of somatic testing to select ovarian cancer cases for germline testing across the board. Please discuss why upfront somatic testing using an assay that includes LGRs is not cost effective (as opposed to using more limited NGS assays selectively in different age groups).
We agree with Reviewer 1 that upfront testing using a tumour BRCA1/2 assay that can detect all pathogenic variants including LGRs is more cost effective that using a limited NGS assay that used in selected age groups. The argument we have attempted to make in our manuscript is that upfront, mainstream germline testing can be reserved for all patients with confirmed tumour BRCA1/2 pathogenic/likely pathogenic variants.
Reviewer 2 Report
This is a retrospective observational study of women with non-mucinous high-grade EOC who had germline and tumor testing for BRCA1/2 and HRR genes/deficiency. There were 48 patients over age 80, and this was the subgroup of interest. In this subgroup, there was only 1 patient with a BRCA germline PV, and that was detected in the tumor. The conclusion is that tumor testing could be done first in this age group, and then confirmatory germline testing if a tumor PV is identified. Patients over age 80 were 9 times more likely to have a somatic vs. germline BRCA 1/2 PV.
Comments for the authors:
1. The rationale for tumor testing first makes sense in the subgroup over age 80, given the high somatic:germline BRCA PV rate. But there was only 1 BRCA germline PV in this subgroup, so it is really valid to conclude that tumor testing is a sufficient triage for confirmatory germline testing given only 1 patient? What about the age group 70-79, in which the somatic: germline BRCA PV rate is still 2:1? In fact, in another publication by the same group (Br J Cancer 2022 Jul;127(1):163-167), the recommendation is that germline testing is not required in women over age 70 if there is no tumor PV. Therefore why focus on the age group over 80?
2. Since the authors had access to Myriad MyChoice CDx testing for large genomic rearrangements, what was the concordance between tumor and germline BRCA testing in this study for all age groups? The age groups with the highest rates of BRCA germline PV are the <50 and 50-59 cohorts, in which the germline BRCA rates were 19% and 20%, respectively. While this rate is sufficiently high to warrant genetic testing, it still means that up to 80% of patients did not have a PV and arguably did not need genetic testing.
3. The authors state in the Discussion (line 196) that “Unselected multi-gene germline and somatic BRCA1/2 testing is currently not cost-effective for accessing PARPi therapy”, but they may reconsider this statement given that there are at least 2 recent publications demonstrating the germline and somatic testing are cost-effective in selecting patients for PARPi , which may not have been available when this manuscript was submitted (Jang et al, Ann Lab Med 2023 Jan 1; 43(1): 73–81; Kwon et al, JCO Precis Oncol 2022 Oct;6:e2200033).
Author Response
Responses are provided in bold
Reviewer 2
This is a retrospective observational study of women with non-mucinous high-grade EOC who had germline and tumor testing for BRCA1/2 and HRR genes/deficiency. There were 48 patients over age 80, and this was the subgroup of interest. In this subgroup, there was only 1 patient with a BRCA germline PV, and that was detected in the tumor. The conclusion is that tumor testing could be done first in this age group, and then confirmatory germline testing if a tumor PV is identified. Patients over age 80 were 9 times more likely to have a somatic vs. germline BRCA 1/2 PV.
Comments for the authors:
- The rationale for tumor testing first makes sense in the subgroup over age 80, given the high somatic:germline BRCA PV rate. But there was only 1 BRCA germline PV in this subgroup, so it is valid to conclude that tumor testing is a sufficient triage for confirmatory germline testing given only 1 patient? What about the age group 70-79, in which the somatic:germline BRCAPV rate is still 2:1? In fact, in another publication by the same group (Br J Cancer 2022 Jul;127(1):163-167), the recommendation is that germline testing is not required in women over age 70 if there is no tumor PV. Therefore, why focus on the age group over 80?
Thank you for your comment and referencing to our Br J Cancer 2022 paper. We agree that the germline BRCA1/2 testing should be reserved for those patients considered high-risk and those with a positive tumour test result. In the Br J Cancer2022 paper we established that patients considered >70 were low risk of germline BRCA1/2 testing. In the latest manuscript we wanted to assess all routine genetic testing assays in low-risk index cases, which is why we included germline BRCA1/2, tumour BRCA1/2, HRD and non-BRCA1/2 HRRm assays. Our findings indicate that tumour BRCA1/2 and HRD testing is only required in this group, as the risk of BRCA1/2-associated familial ovarian cancer is very low.
- Since the authors had access to Myriad MyChoice CDx testing for large genomic rearrangements, what was the concordance between tumor and germline BRCA testing in this study for all age groups? The age groups with the highest rates of BRCA germline PV are the <50 and 50-59 cohorts, in which the germline BRCA rates were 19% and 20%, respectively. While this rate is sufficiently high to warrant genetic testing, it still means that up to 80% of patients did not have a PV and arguably did not need genetic testing.
The concordance between pathogenic BRCA1/2 variants in germline and tumour DNA using Myriad’s myChoice CDx was 100%. All germline LGRs were detected in tumour DNA using Myriad’s myChoice CDx. We agree with Reviewer 2 that there are many women who unnecessarily undergo germline BRCA1/2 testing. The inclusion of PARPi as standard therapy to treat high-grade epithelial ovarian cancer has led to mainstream testing in unselected cases. Testing tumour DNA using a NGS assay that can detect all types of BRCA1/2 variants (e.g., the Myriad myChoice® CDx) would ideally be used upfront in all cases and then germline BRCA1/2 testing performed thereafter in only tBRCA+ cases. This strategy would avoid unnecessary germline BRCA1/2 testing. This is likely to be the future direction of molecular testing in epithelial ovarian cancer.
- The authors state in the Discussion (line 196) that “Unselected multi-gene germline and somatic BRCA1/2testing is currently not cost-effective for accessing PARPi therapy”, but they may reconsider this statement given that there are at least 2 recent publications demonstrating the germline and somatic testing are cost-effective in selecting patients for PARPi, which may not have been available when this manuscript was submitted (Jang et al, Ann Lab Med 2023 Jan 1; 43(1): 73–81; Kwon et al, JCO Precis Oncol 2022 Oct;6:e2200033).
Thank you for your suggestion. We were unaware of these publications at the time of submission. We have amended the sentence identified by Reviewer 2 to account for these papers. We have also cited both references in the manuscript now.
Round 2
Reviewer 2 Report
Thank you for responding to the queries. However the statement about concordance between tumor and germline BRCA testing being 100% for all age groups cannot be found in the revised manuscript, as indicated in the response to the initial review. The readers could figure this out from the tables, but it is still important to highlight. Maybe this is already in the revision but missed.
Author Response
Responses are provided in bold
Reviewer 2
Thank you for responding to the queries. However, the statement about concordance between tumour and germline BRCA testing being 100% for all age groups cannot be found in the revised manuscript, as indicated in the response to the initial review. The readers could figure this out from the tables, but it is still important to highlight. Maybe this is already in the revision but missed.
We have now included this data now.